# UNDERSTANDING KNOWLEDGE DISTILLATION IN NON-AUTOREGRESSIVE MACHINE TRANSLATION

**Chunting Zhou**[1]*,**Jiatao Gu**[2]*, **Graham Neubig**[1]
Language Technologies Institute, Carnegie Mellon University[1]
Facebook AI Research[2]
{chuntinz, gneubig}@cs.cmu.edu, jgu@fb.com

## ABSTRACT

Non-autoregressive machine translation (NAT) systems predict a sequence of output tokens in parallel, achieving substantial improvements in generation speed compared to autoregressive models. Existing NAT models usually rely on the technique of knowledge distillation, which creates the training data from a pretrained autoregressive model for better performance. Knowledge distillation is empirically useful, leading to large gains in accuracy for NAT models, but the reason for this success has, as of yet, been unclear. In this paper, we first design systematic experiments to investigate why knowledge distillation is crucial in NAT training. We find that knowledge distillation can reduce the complexity of data sets and help NAT to model the variations in the output data. Furthermore, a strong correlation is observed between the capacity of an NAT model and the complexity of the distilled data that provides the best translation quality. Based on these findings, we further propose several approaches that can alter the complexity of data sets to improve the performance of NAT models. We achieve state-of-the-art performance for NAT-based models, and close the gap with the autoregressive baseline on the WMT14 En-De benchmark.[1]

## 1 INTRODUCTION

Traditional neural machine translation (NMT) systems (Bahdanau et al., 2015; Gehring et al., 2017; Vaswani et al., 2017) generate sequences in an autoregressive fashion; each target token is predicted step-by-step by conditioning on the previous generated tokens in a monotonic (e.g. left-to-right) order. While such autoregressive translation (AT) models have proven successful, the sequential dependence of decisions precludes taking full advantage of parallelism afforded by modern hardware (e.g. GPUs) at inference time. In contrast, non-autoregressive translation (NAT) models (Gu et al., 2018; Lee et al., 2018) predict the whole sequence or multi-token chunks of the sequence simultaneously, alleviating this problem by trading the model's capacity for decoding efficiency. Such a non-autoregressive factorization assumes that the output tokens are independent from each other. However, this assumption obviously does not hold in reality and as a result NAT models generally perform worse than standard AT models.

One key ingredient in the training recipe for NAT models that is used in almost all existing works (Gu et al. (2018); Lee et al. (2018); Stern et al. (2019), *inter alia*) is creation of training data through *knowledge distillation* (Hinton et al., 2015). More precisely, *sequence-level knowledge distillation* (Kim & Rush, 2016) – a special variant of the original approach – is applied during NAT model training by replacing the target side of training samples with the outputs from a pre-trained AT model trained on the same corpus with a roughly equal number of parameters. It is usually assumed (Gu et al., 2018) that knowledge distillation's reduction of the "modes" (alternative translations for an input) in the training data is the key reason why distillation benefits NAT training. However, this intuition has not been rigorously tested, leading to three important open questions:

---

*Equal Contribution. Most work was done during Chunting's internship at FAIR.

[1]Code is released at https://github.com/pytorch/fairseq/tree/master/examples/nonautoregressive_translation.

- Exactly how does distillation reduce the "modes", and how we could we measure this reduction quantitatively? Why does this reduction consistently improve NAT models?
- What is the relationship between the NAT model (student) and the AT model (teacher)? Are different varieties of distilled data better for different NAT models?
- Due to distillation, the performance of NAT models is largely bounded by the choice of AT teacher. Is there a way to further close the performance gap with standard AT models?

In this paper, we aim to answer the three questions above, improving understanding of knowledge distillation through empirical analysis over a variety of AT and NAT models. Specifically, our contributions are as follows:

- We first visualize explicitly on a synthetic dataset how modes are reduced by distillation (§3.1). Inspired by the synthetic experiments, we further propose metrics for measuring complexity and faithfulness for a given training set. Specifically, our metrics are the *conditional entropy* and *KL-divergence* of word translation based on an external alignment tool, and we show that these metrics are correlated with NAT model performance (§3.2).
- We conduct a systematic analysis (§4) over four AT teacher models and six NAT student models with various architectures on the standard WMT14 English-German translation benchmark. These experiments find a strong correlation between the capacity of an NAT model and the optimal dataset complexity that results in the best translation quality.
- Inspired by these observations, we propose approaches to further adjust the complexity of the distilled data in order to match the model's capacity (§5). We also show that we can achieve the state-of-the-art performance for NAT models and largely match the performance of the AT model.

## 2 BACKGROUND

### 2.1 NON-AUTOREGRESSIVE NEURAL MACHINE TRANSLATION

In order to model the joint probability of the output sequence $y$, NMT models usually generate each output token conditioned on the previously generated ones $p(y|x) = \prod_{t=1}^{T} p(y_t|y_{<t}, x)$. This is known as the *autoregressive factorization*. To generate a translation from this model, one could predict one token at a time from left to right and greedily take $\arg\max$ over each output probability distribution, or use beam search to consider a fixed number of hypotheses. In this work, we study non-autoregressive translation (NAT), a special subset of NMT models with an additional restriction (the zeroth-order Markov assumption) upon the output predictions or a subset thereof. The simplest formulation of an NAT model independently factors the conditional distribution: $p(y|x) = \prod_{t=1}^{T} p(y_t|x)$.

Standard NAT models (Gu et al., 2018) adopt an architecture similar to the Transformer (Vaswani et al., 2017) and make non-autoregressive predictions for the entire sequence with **one** forward pass of the decoder. However, because multiple translations are possible for a single input sentence (the so-called multi-modality problem; Gu et al. (2018)), vanilla NAT models can fail to capture the dependencies between output tokens. As a result, they tend to make egregious mistakes such as outputting tokens repeatedly. To improve the model's ability to handle multi-modality, recent works have incorporated approaches including (1) relaxing the fully non-autoregressive restriction and adopting $K$ decoding passes (instead of just one) to iteratively refine the generated outputs (Lee et al., 2018; Ghazvininejad et al., 2019; Wang et al., 2018; Stern et al., 2018; 2019; Gu et al., 2019); (2) using latent variables (Kaiser et al., 2018; Ma et al., 2019; Shu et al., 2019) or structured information such as syntax trees (Akoury et al., 2019) to capture translation variation; (3) training NAT models with objectives other than maximum likelihood (Wang et al., 2019; Wei et al., 2019; Shao et al., 2019) which ameliorates the effects of multi-modality. However, to achieve competitive performance with the autoregressive model, almost all existing NAT models rely on training using data distilled from a pre-trained AT model instead of the real parallel training set, as described below.

### 2.2 SEQUENCE-LEVEL KNOWLEDGE DISTILLATION

Knowledge distillation (Liang et al., 2008; Hinton et al., 2015) was originally proposed for training a weaker student classifier on the targets predicted from a stronger teacher model. A typ-

ical approach is using the label probabilities produced by the teacher as "soft targets" $q_i = \exp(z_i/\tau)/\sum_j \exp(z_j/\tau)$ for training the student model, where $q_i$ and $z_i$ are the probability and the logit of class $i$ respectively and $\tau$ is the temperature. Prior work has shown the effectiveness of adopting knowledge distillation in adversarial defense (Papernot et al., 2016), neural network compression (Howard et al., 2017), and fast inference for speech synthesis (Oord et al., 2018).

In the context of sequence generation, Kim & Rush (2016) extend knowledge distillation to the sentence level using "hard targets" from a pretrained large teacher model to train a small sequence generation model. More precisely, the teacher distribution $q(\boldsymbol{t}|\boldsymbol{x})$ is approximated by its mode: $q(\boldsymbol{t}|\boldsymbol{x}) \approx \mathbb{1}\{\boldsymbol{t} = \arg\max_{\boldsymbol{t} \in \mathcal{T}} q(\boldsymbol{t}|\boldsymbol{x})\}$ with the following objectives:

$$\mathcal{L}_{\text{seq-KD}} = -\mathbb{E}_{\boldsymbol{x} \sim \text{data}} \sum_{\boldsymbol{t} \in \mathcal{T}} q(\boldsymbol{t}|\boldsymbol{x}) \log p(\boldsymbol{t}|\boldsymbol{x}) \approx -\mathbb{E}_{\boldsymbol{x} \sim \text{data}, \hat{\boldsymbol{y}} = \arg\max_{\boldsymbol{t} \in \mathcal{T}} q(\boldsymbol{t}|\boldsymbol{x})} \left[\log p(\boldsymbol{t} = \hat{\boldsymbol{y}}|\boldsymbol{x})\right], \quad (1)$$

where $\boldsymbol{t} \in \mathcal{T}$ is the space of possible target sequences. This can also be seen as a special case of standard distillation over the sentence space when the temperature $\tau$ approaches 0, which is equivalent to taking the $\arg\max$ over all feasible translations. While the "hard target" $\hat{\boldsymbol{y}}$ is the most likely translation predicted by the teacher, in practice we use beam search as an approximation. As mentioned earlier, almost all the existing literature trains NAT models using sequence-level knowledge distillation from a pre-trained AT model to achieve competitive performance. Particularly, it is common to train the teacher model as a standard autoregressive Transformer (Vaswani et al., 2017) with a roughly equal number of trainable parameters as the desired NAT model on the real data. Next, we will first study how this knowledge distillation process affects the behavior of NAT models.

## 3 How does Distillation Improve NAT?

In this section, we start from an introductory example to illustrate how NAT models fail to capture the multi-modality of data. Then we propose a metric to assess the multi-modality of a data set and use it to test our hypothesis about how knowledge distillation affects NAT models.

### 3.1 Synthetic Experiment for Multi-modality

**Dataset.** We start by investigating NAT's difficulties in modeling multi-modality in output data using a synthetic setup where we explicitly include multiple modes in the training data. More specifically, we utilize three language pairs – English-German (En-De), English-French (En-Fr), and English-Spanish (En-Es) – from the Europarl parallel corpus.[2] We extract sentences that have aligned sentences for all languages, and create a multi-target En-De/Es/Fr corpus. In this case every English input sentence always corresponds to target sentences in three different languages, which forms three *explicit* output modes. Notably, this is similar to the one-to-many translation setting in Johnson et al. (2017) but in our case we do not have an explicit signal (e.g. target language tag) to tell the NMT model which target language to translate to.

**Models.** We train both the AT and NAT models on this concatenated data set, then compare the distributions of translations with each other. We use the standard Transformer(base) model (Vaswani et al., 2017) as the AT model, and a simplified version of Gu et al. (2018) as the NAT model where the decoder's inputs are monotonically copied from the encoder embeddings and a length predictor is learned to predict the target sentence length. Both models are trained for $300,000$ steps using maximum likelihood. After training, we use both models to translate the English sentences in the validation and test sets.

**Visualization of AT Outputs.** The synthetic setup enables us to better understand and visualize the modes in the outputs more easily. First, we visualize the outputs from the AT model. For every translated sentence, we visualize the estimated probability distribution of language classes as a point in Fig. 1 (a). This probability is calculated as the average of the posterior probability of each token, and it is estimated based on the Bayes' law:

$$p(l_i|\boldsymbol{y}) \approx \frac{1}{T} \sum_{t=1}^{T} p(l_i|y_t) = \frac{1}{T} \sum_{t=1}^{T} \frac{p(y_t|l_i)p(l_i)}{\sum_k p(y_t|l_k)p(l_k)} \quad (2)$$

---

[2]https://www.statmt.org/europarl/

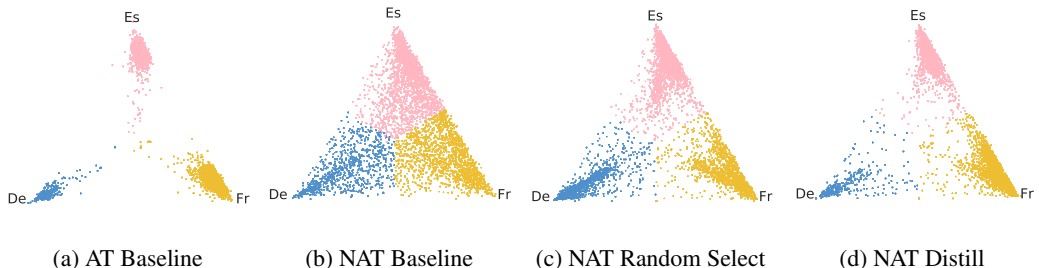

| (a) AT Baseline | (b) NAT Baseline | (c) NAT Random Select | (d) NAT Distill |

Figure 1: Posterior distribution of language IDs for the outputs from different models. Each translation is represented as a point inside the simplex $\Delta^2 = \{(p_{de}, p_{es}, p_{fr})|p_k \in (0,1), p_{de}+p_{es}+p_{fr} = 1\}$ where $p_k$ is the estimated probability of being translated into language $k \in (de, es, fr)$. We distinguish the language that has the largest probability with different colors.

where $l_i$ denotes the language class $i$, and $p(y_t|l_i)$ is the token frequency of $y_t$ in language $l_i$. We assume $p(l_i)$ follows a uniform distribution. As shown in Fig. 1 (a), points of the AT outputs are clustered closely to each vertex of the simplex, indicating that the AT model prefers to generate the whole sequence in one language. This phenomenon verifies our assumption that decoding with the AT model (distillation) is essentially selecting "modes" over the real data.

**Visualization of NAT Outputs.** We visualize outputs for the NAT model trained on the same data in Fig. 1 (b). In contrast to the AT results, the NAT points are scattered broadly inside the simplex, indicating that the NAT model fails to capture the mode of language types. Instead, it predicts tokens mixed with multiple languages, which corroborates our hypothesis that the NAT model has trouble consistently selecting a single mode when multiple modes exist.

Next, we create two datasets that have fewer modes than the original dataset. First, we randomly select a single target sentence from one of the three languages for each source sentence. Second, we perform distillation, decoding from the AT model trained on the combined training set. As noted in the AT results, distillation will also roughly be selecting a language mode, but we conjecture that this selection may be more systematic, selecting a particular language for a particular type of training sentence. As shown in Fig. 1(c) (d), NAT models trained on both of these datasets are more likely to choose one mode (language) when generating translations, showing that training with reduced modes is essential for NAT model. Furthermore, points in Fig. 1 (d) are clearly clustered better than (c) indicating that modes selected by AT models are indeed likely more systematic and easy to capture than those generated by randomly assigning a language for each sentence.

## 3.2  QUANTITATIVE MEASURES FOR PARALLEL DATA

To better study why distillation is crucial for NAT models, in this section, we propose quantitative measures for analyzing the *complexity* and *faithfulness* of parallel data, two properties that we hypothesize are important for NAT training.

**Measure of Complexity.** Inspired by the observations in the synthetic experiments, we propose to use a measure of translation *uncertainty*, specifically operationalized as conditional entropy, as the measurement of complexity $C(d)$ for any given dataset $d = \{(\boldsymbol{x}_1, \boldsymbol{y}_1), ..., (\boldsymbol{x}_N, \boldsymbol{y}_N)\}$, where $(\boldsymbol{x}, \boldsymbol{y})$ is sentence pair instantiation of $(\mathbf{X}, \mathbf{Y})$ and $\mathbf{X} \in \mathcal{X}, \mathbf{Y} \in \mathcal{Y}$:

$$
\begin{aligned}
\mathcal{H}(\mathbf{Y}|\mathbf{X} = \boldsymbol{x}) &= \sum_{\boldsymbol{y} \in \mathcal{Y}} p(\boldsymbol{y}|\boldsymbol{x}) \log p(\boldsymbol{y}|\boldsymbol{x}) \\
&\approx \sum_{\boldsymbol{y} \in \mathcal{Y}} (\prod_{t=1}^{T_y} p(y_t|\boldsymbol{x}))(\sum_{t=1}^{T_y} \log p(y_t|\boldsymbol{x})) \qquad \text{asm.1: conditional independence} \\
&\approx \sum_{t=1}^{T_y} \sum_{y_t \in \mathcal{A}(\boldsymbol{x})} p(y_t|\text{Align}(y_t)) \log p(y_t|\text{Align}(y_t)) \qquad \text{asm.2: alignment model} \\
&= \sum_{t=1}^{T_x} \mathcal{H}(y|x = x_t)
\end{aligned}
\tag{3}
$$

| $d$ | En-De | En-Es | En-Fr | Full Real Data | Random Selection | Distillation |
|---|---|---|---|---|---|---|
| $C(d)$ | 3.12 | 2.81 | 2.89 | 3.67 | 3.30 | **2.64** |

Table 1: Complexity $C(d)$ (↑ more complex) of the Europarl data set of different settings in §3.1.

where we use $x$ and $y$ to denote a word in the source and target vocabulary respectively. $T_x$ and $T_y$ denote the length of the source and target sentences. To make the computation tractable, we make two additional assumptions on the conditional distribution $p(\boldsymbol{y}|\boldsymbol{x})$:

- *Assumption* 1: We assume the target tokens are independent given the source sentence. Then the conditional entropy of a sentence can be converted into the sum of entropy of target words conditioned on the source sentence $\boldsymbol{x}$.

- *Assumption* 2: We assume the distribution of $p(y_t|\boldsymbol{x})$ follows an alignment model (Dyer et al., 2013)[3] where $y_t$ is is generated from the word alignment distribution $p(y_t|\mathrm{Align}(y_t))$. This makes it possible to simplify the conditional entropy to the sum of entropy of target words conditioned on the aligned source words denoted $\mathcal{H}(y|x = x_t)$.

The corpus level *complexity* $C(d)$ is then calculated by adding up the conditional entropy $\mathcal{H}(\mathbf{Y}|\mathbf{X} = \boldsymbol{x})$ of all sentences. To prevent $C(d)$ from being dominated by frequent words, we calculate $C(d)$ by averaging the entropy of target words conditioned on a source word, denoted $C(d) = \frac{1}{|\mathcal{V}_x|} \sum_{x \in \mathcal{V}_x} \mathcal{H}(y|x)$.

To illustrate that the proposed metric is a reasonable measure of complexity of a parallel corpus, in Tab. 1 we compute $C(d)$ for parallel data from different language pairs, the concatenated data set, and the data distilled from the AT model described in §3.1. We observe that the conditional entropy of the distilled data is much smaller than that of the concatenated or randomly selected data mentioned above. Additionally, we find that the conditional entropy of En-Es and En-Fr are similar but that of En-De is relatively larger, which can also explain why the student NAT model prefers to predict the modes of Es or Fr more often than De as shown in Fig. 1(d).

**Measure of Faithfulness.** $C(d)$ reflects the level of multi-modality of a parallel corpus, and we have shown that a simpler data set is favorable to an NAT model. However, it is not fair to assess the data set only by its complexity; we can trivially construct a simple data set with no variations in the output, which obviously won't be useful for training. The other important measurement of the data set is its *faithfulness* to the real data distribution. To measure the faithfulness of a parallel corpus $d$, we use KL-divergence of the alignment distribution between the real parallel data set $r$ and an altered parallel data set $d$, denoted $F(d)$:

$$F(d) = \frac{1}{|\mathcal{V}_x|} \sum_{x \in \mathcal{V}_x} \sum_{y \in \mathcal{V}_y} p_r(y|x) \log \frac{p_r(y|x)}{p_d(y|x)} \tag{4}$$

## 4 EMPIRICAL STUDY

In this section, we perform an extensive study over a variety of non-autoregressive (NAT) models trained from different autoregressive (AT) teacher models to assess how knowledge distillation affects the performance of NAT models.

### 4.1 EXPERIMENTAL SETTINGS

**Data.** We use the data set commonly used by prior work as our evaluation benchmark: WMT14 English-German (En-De)[4]. We use `newstest2013` as the validation set for selecting the best model, and `newstest2014` as the test set. We learn a byte-pair encoding (BPE, Sennrich et al., 2016) vocabulary of 37,000 on the tokenized data.

**AT Models.** We set up four Transformer models with different parameter sizes: Transformer-tiny/small/base/big denoted as *tiny*, *small*, *base*, *big* respectively. We build *base* and *big* models

---
[3]We follow `https://github.com/clab/fast_align` to compute the alignment given the dataset.
[4]`http://www.statmt.org/wmt14/translation-task.html`

following settings described in Vaswani et al. (2017), and reduce the model sizes for *tiny*, *small* to create weaker teacher models. Details of the model architectures can be found in Appendix A.

All the models are trained using the Adam optimizer (Kingma & Ba, 2014) with the maximum number of steps set to $300,000$. After training, we use the resulting AT models to decode the whole training set with beam size $5$ and replace the real target sentences to create a new parallel corpus.

**NAT Models.** We consider the following NAT models, from vanilla to state-of-the-art. All the models are using the Transformer as the basic backbone and are (re-)implemented based on Fairseq[5] except for FlowSeq. We briefly outline the methods and parameters here, and describe detailed settings in the Appendix A.

- **Vanilla NAT (Gu et al., 2018):** Similarly to §3.1, we use a simplified version where the decoder's inputs are directly copied from the encoder without considering latent variables.

- **FlowSeq (Ma et al., 2019):** FlowSeq adopts normalizing flows (Kingma & Dhariwal, 2018) as the latent variables to model the mappings from source sentences to a latent space.

- **NAT with Iterative Refinement (iNAT, Lee et al., 2018):** iNAT extends the vanilla NAT by iteratively reading and refining the translation. The number of iterations is set to 10 for decoding.

- **Insertion Transformer (InsT, Stern et al., 2019):** InsT adopts a similar architecture as iNAT while generating the sequence by parallel insertion operations. Here, we only consider InsT trained with uniform loss as described in the original paper.

- **MaskPredict (MaskT, Ghazvininejad et al., 2019):** MaskT adopts a masked language model (Devlin et al., 2018) to progressively generate the sequence from an entirely masked input. The number of iterations is set to be 10.

- **Levenshtein Transformer (LevT, Gu et al., 2019):** LevT uses similar architectures as in InsT and MaskT while generating based on both insertion and deletion operations. We experiment with a base and big LevT model (LevT and LevT-big in Tab. 2).

We also summarize the parameter size, performance and relative decoding speed of the NAT models introduced in Tab. 2. We use the decoding time of vanilla NAT to represent one unit of time, and `Iters × Pass` represents the relative time units used for each model.

As mentioned earlier, we analyze each model by training from both the real and $4$ distilled targets. We train the NAT models for the same number of steps as the AT models. For a fair comparison of the actual ability of each NAT-based model, we test all the models based on greedy decoding without any advanced search algorithms (e.g. length beam (Ghazvininejad et al., 2019), noisy parallel decoding (Ma et al., 2019), or re-ranking from the teacher model (Gu et al., 2018)). Notably, the vanilla NAT and FlowSeq output translations with single forward pass, while the remaining models are based on the iterative refinement.

| Models | Params | BLEU | Pass | Iters |
|---|---|---|---|---|
| AT models | | | | |
| AT-tiny | 16M | 23.3 | – | $n$ |
| AT-small | 37M | 25.6 | – | $n$ |
| AT-base | 65M | 27.1 | – | $n$ |
| AT-big | 218M | 28.2 | – | $n$ |
| NAT models | | | | |
| vanilla | 71M | 11.4 | 1 | 1 |
| FlowSeq | 73M | 18.6 | 13 | 1 |
| iNAT | 66M | 19.3 | 1 | $k \ll n$ |
| InsT | 66M | 20.9 | 1 | $\approx \log_2 n$ |
| MaskT | 66M | 23.5 | 1 | 10 |
| LevT | 66M | 25.2 | 1 | $3k \ll n$ |
| LevT-big | 220M | 26.5 | $\approx 3$ | $3k \ll n$ |

Table 2: AT and NAT models. Number of parameters and test BLEU when trained on the real data demonstrate model capacity. `Iters` is number of passes used in decoding for output length $n$ and hyperparameter $k$. `Pass` is relative time used for one pass of decoding.

### 4.2 Analysis of the Distilled Data

We compare different dimensions of the data generated by the four AT models and the real data set in Fig. 3. First, Fig. 3 (a) shows that as the capacity of the AT model increases, the complexity $C(d)$ of the distilled data increases, which indicates that the multi-modality increases as well. At the same time, we observe that $F(d)$ defined in §3.2 also decreases, showing that the distilled data more faithfully represents the word-level translation distribution of the original data.

---

[5]https://github.com/pytorch/fairseq

| | |
|---|---|
| Source | For more than 30 years , Josef Winkler has been writing from the heart , telling of the hardships of his childhood and youth . |
| Distilled Target | Seit mehr als 30 Jahren schreibt Josef Winkler aus dem Herzen und erzählt von der Not seiner Kindheit und Jugend . |
| Real Target | Josef Winkler schreibt sich seit mehr als 30 Jahren die Nöte seiner Kindheit und Jugend von der Seele . |

Figure 2: A sampled pair together with its real target from the distilled data of the base-AT model. Chunks annotated in the same colors are approximately aligned with each other.

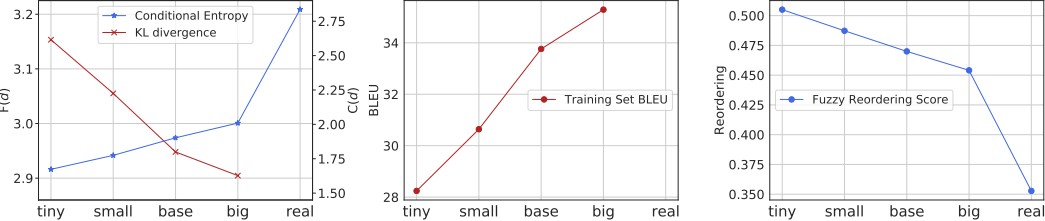

Figure 3: Complexity $C(d)$ (↑ more complex), faithfulness $F(d)$ (↓ more faithful), training BLEU, and reordering score (↑ more monotonic alignment) of different distilled sets of WMT14-ENDE.

Second, we plot the BLEU score of the distilled data w.r.t to the real data set in (b) and we observe that the BLEU score of the distilled data from a higher-capacity teacher model is higher, which is both intuitive and in agreement with the results on KL divergence.

We also investigate how the relative ordering of words in the source and target sentences is changed during distillation. We use the fuzzy reordering score proposed in Talbot et al. (2011). A larger fuzzy reordering score indicates the more monotonic alignments. As shown in Fig 3 (c), the distilled data has significantly less reordering compared to the real parallel sentences, and the distilled data from a weaker AT teacher is more monotonic than a stronger AT teacher. We also show a randomly sampled example in Fig. 2 where compared to the real translation, the AT distilled target is much more monotonically aligned to the source sentence. This has potential benefits in that these simpler reordering patterns may be easier to learn for NAT models, but also disadvantages in that it may prevent NAT models from learning complex reordering patterns.

### 4.3 ANALYSIS OF DISTILLATION STRATEGIES

In §4.2, we have shown that decoding with an AT model reduces the conditional entropy of the parallel data set, which mitigates multi-modality in the output data. But does the decoding method of the AT model affect this change in the data set? We also investigate different decoding strategies when creating distilled data, using the base Transformer model as the teacher and the vanilla NAT model as the student. In Tab. 3, four decoding methods are presented: sampling, sampling within the top-10 candidates, beam search, and greedy decoding. With the same AT model, the performance of the NAT model differs widely depending on the decoding approach, where distillation with beam search results in the best performance.

We can see that beam search or greedy decoding can reduce the complexity of the real data the most while maintaining high faithfulness. In contrast, sampling based decoding methods less aggressively reduce the modes in the output sequence. This finding is in concert with Ott et al. (2018), who demonstrate that because beam search approximately selects the most probable translation, it effectively reduces diversity in the output translations compared to sampling or the true distribution.

| Decoding Method | $C(d)$ | $F(d)$ | BLEU |
|---|---|---|---|
| sampling | 3.623 | 3.354 | 6.6 |
| sampling (Top 10) | 2.411 | 2.932 | 14.6 |
| greedy | 1.960 | 2.959 | 18.9 |
| beam search | 1.902 | 2.948 | **19.5** |

Table 3: Comparisons of decoding methods on WMT14-ENDE newstest 2014 test set.

### 4.4 DISTILLED DATA V.S. NAT MODELS

We next examine the relationship between the NAT students and distilled training data from different AT models. In Fig. 4, we demonstrate results for the NAT models listed in §4.1. We use the test

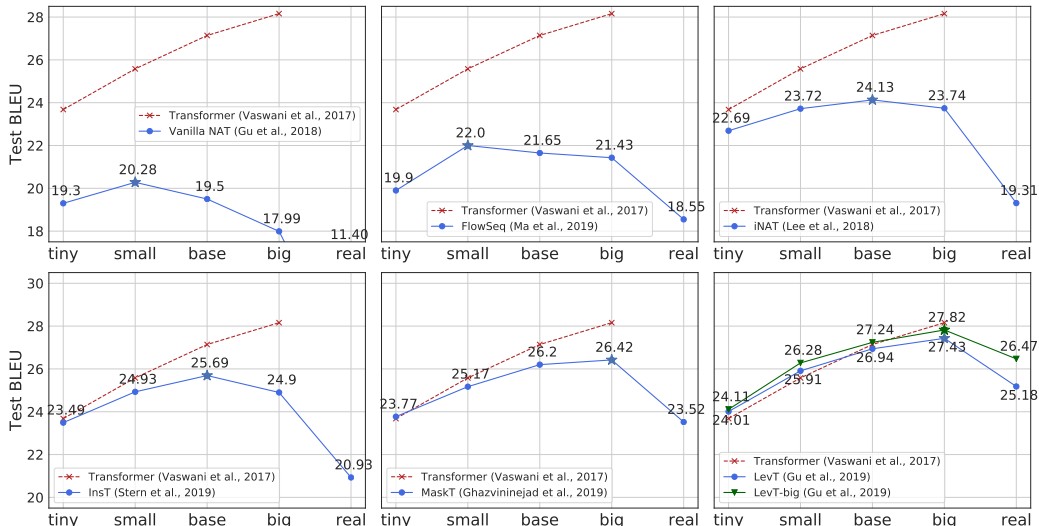

Figure 4: The performance of NAT models of varying capacity trained on both the real and the distilled data from tiny, small, base and big AT models on WMT14-ENDE newstest 2014 test sets.

set performance on real data as a simple metric to measure the capacity of the NAT model and arrange the subfigures in an increasing order of the performance (left-to-right, top-to-bottom). The results in the figure demonstrate that, interestingly, weaker NAT students prefer distilled data with smaller complexity as measured above in §4.2. The best performance of NAT models – from lower capacity ones to higher capacity ones – is achieved with distilled data of lower complexity to higher complexity, i.e. the vanilla NAT model performs best when using the distilled data from a small Transformer whereas LevT achieves the best performance when training with the distilled data from a big Transformer. Third, and notably, by simply changing the distilled data set upon which the models are trained, we are able to significantly improve the state-of-the-art results for models in a particular class. For example, FlowSeq increased to 22, by simply changing from the distilled data of Transformer(base) to Transformer(small). Finally, we find that by distilling from a big AT model, LevT is able to close the gap with the Transformer (base) with a similar number of parameters. Both LevT and LevT-big achieve the state-of-the-art performance for NAT-based models.

## 5 IMPROVEMENTS TO KNOWLEDGE DISTILLATION

The previous section shows that the optimal complexity of the dataset is highly correlated with the capacity of the NAT model. In this section, we introduce three techniques that can be used to alter the distilled data to match the capacity of NAT model. Specifically, these techniques can be used to simplify the data further (BANs, MoE) for a lower-capacity student model or increase faithfulness of the data set (Interpolation) for a higher-capacity student model.

**Born-Again Networks.** We apply Born-Again neworks (BANs) to create a simplified dataset for NAT models. BANs were originally proposed as a self-distillation technique (Furlanello et al., 2018) that uses the output distribution of a trained model to train the original model. Starting from the real data, we repeatedly train new AT models with decoded sentences from the AT model at the previous iteration. This process is repeated for $k$ times and yields $k$ distilled data sets, upon which we perform NAT training and examine how the $k$ born-again teachers affect the performance of NAT students.

We conduct experiments using the vanilla NAT model (Gu et al., 2018) (which achieved the best performance with distilled data from a small Transformer in §4.4) and the base Transformer as the AT model. As shown in Fig. 5, we can make the following observations: (i) The performance of the base AT model almost remains unchanged during the reborn iterations. (ii) The performance of the vanilla NAT model can be improved by 2 BLEU when using the distilled data from reborn iteration 6. (iii) As the reborn iterations continue, the complexity of the distilled data decreases and becomes constant eventually. Meanwhile, the quality of the distilled data compared to the real data decreases.

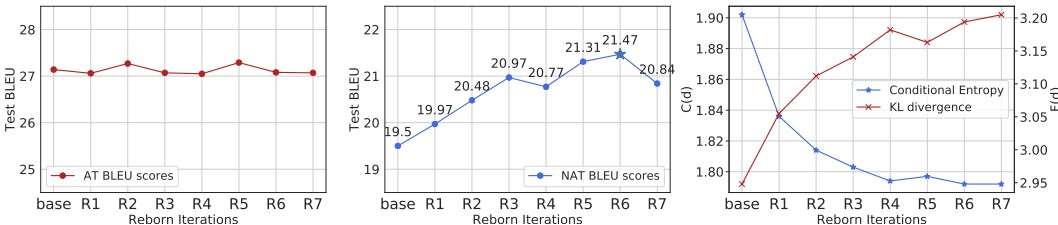

Figure 5: Reborn experiments: (from left to right) performance of the base AT model, performance of the vanilla NAT model, $C(d)$ and $F(d)$ of distilled data sets. R-$i$ denotes the $i$-th reborn iteration.

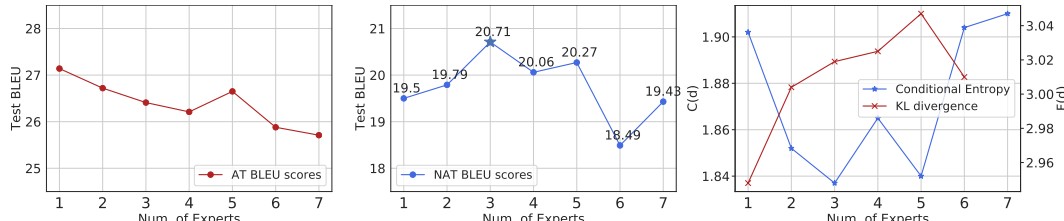

Figure 6: MoE experiments: (from left to right) performance of the base AT model, performance of the vanilla NAT model, $C(d)$ and $F(d)$ of distilled data sets w.r.t the number of experts.

**Mixture-of-Experts.** The mixture-of-expert model (MoE; Shen et al. (2019)) learns different experts for diverse machine translation, and different mixture components were shown to capture consistent translation styles across examples. Inspired by this, we use one expert from the mixture model to translate the training data, which is supposed to generate a single style of translation and reduce the diversity in the original data set. Then we use the best single-expert translations as the distilled data to train the vanilla NAT model. Specifically, we follow Shen et al. (2019)'s setup, using the base Transformer model and uniform hard mixture model, varying the number of experts.

In Fig. 6, we observe that the performance of the best expert of MoE tends to decrease as the number of experts increases. However, the complexity ($C(d)$) and faithfulness ($F(D)$) of distilled data from different MoE models has a relatively large variance. Compared to using the distilled data from a plain base AT model, the performance of NAT model is improved by 1.21 BLEU when using the distilled data from the MoE model with the number of experts of 3 which produces the distilled data with the least complexity.

**Sequence-Level Interpolation.** §4.4 shows stronger NAT models (e.g. MaskT, LevT) have the ability to learn from the dataset that is closer to the real data, and achieve better performance. We adopt the sequence-level interpolation proposed in Kim & Rush (2016) as a natural way to create a better dataset. Different from distillation, interpolation picks the sentence with the highest sentence-level BLEU score w.r.t. the ground truth from $K-$best beam search hy-

| $d$ | $C(d)$ | $F(d)$ | BLEU |
|---|---|---|---|
| base | 1.902 | 2.948 | 26.94 |
| base-inter | 1.908 | 2.916 | **27.32** |

Table 4: Results w/ and w/o sequence-level interpolation with LevT.

potheses. In our experiments, we first run beam search using the base Transformer model with a beam size of 5 then select the sentences with the highest BLEU score from the top-3 candidates.

Tab. 4 compares the performance of LevT trained with distilled data from the AT model with the standard distillation or interpolation. We observe that selection with BLEU score from the base AT model (base-inter) improves the performance of LevT $\sim 0.4$ BLEU while the dataset complexity $C(d)$ does not increase much.

# 6 CONCLUSION

In this paper, we first systematically examine why knowledge distillation improves the performance of NAT models. We conducted extensive experiments with autoregressive teacher models of different capacity and a wide range of NAT models. Furthermore, we defined metrics that can quantitatively measure the complexity of a parallel data set. Empirically, we find that a higher-capacity

NAT model requires a more complex distilled data to achieve better performance. Accordingly, we propose several techniques that can adjust the complexity of a data set to match the capacity of an NAT model for better performance.

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

## A  Experimental Details

### A.1  AT Models

**Model**  All the AT models are implemented based on the Transformer model using fairseq (Ott et al., 2019), and we basically follow the fairseq examples to train the transformers[6]. Following the notation from Vaswani et al. (2017), we list the basic parameters of all the AT model we used:

| Models | *tiny* | *small* | *base* | *big* |
|---|---|---|---|---|
| $d_{\mathrm{model}}$ | 256 | 512 | 512 | 1024 |
| $d_{\mathrm{hidden}}$ | 1024 | 1024 | 2048 | 4096 |
| $n_{\mathrm{layers}}$ | 3 | 3 | 6 | 6 |
| $n_{\mathrm{heads}}$ | 4 | 8 | 8 | 16 |
| $p_{\mathrm{dropout}}$ | 0.1 | 0.1 | 0.3 | 0.3 |

Table 5: Basic hyper-parameters of architecture for AT models.

**Training**  For all experiments, we adopt the Adam optimizer (Kingma & Ba, 2014) using $\beta_1 = 0.9, \beta_2 = 0.98, \epsilon = 1e - 8$. The learning rate is scheduled using `inverse_sqrt` with a maximum learning rate $0.0005$ and $4000$ warmup steps. We set the label smoothing as $0.1$. All the models are run on $8$ GPUs for $300,000$ updates with an effective batch size of $32,000$ tokens. The best model is selected based on the validation loss except for FlowSeq which uses valid BLEU score.

**Decoding**  After training, we use beam-search with a fixed beam size $5$ for all AT models to create the distilled dataset. We use length normalization without length penalty.

### A.2  NAT Models

**Model**  Tab. 2 also lists all the NAT models we test in this work. In general, all the NAT models except FlowSeq and LevT-big adopts a similar architecture and hyper-parameters as the Transformer-base (see Tab. 5). LevT-big is a naive extension of the original LevT model with a comparable parameter setting as Transformer-big (Tab. 5). For FlowSeq, we use the base model (FlowSeq-base) described in (Ma et al., 2019). We re-implemented the vanilla NAT as a simplified version of Gu et al. (2018) where instead of modeling fertility as described in the original paper, we monotonically copy the encoder embeddings to the input of the decoder. All the models except InsT require the additional module to predict the length of the output sequence, or the number of placeholders to be inserted, which is implemented as a standard softmax classifier over the lengths of [0, 256). For LevT, we also have a binary classifier to predict the deletion of the incorrect tokens.

**Training**  Similar to the AT models, all the NAT models are trained using the Adam optimizer with the same learning rate scheduler, in which the warmup steps are set to $10,000$. We train the FlowSeq model on 32 GPUs with a batch size as 2048 sentences, while all the other models are trained on 8 GPUs with an effective batch size of $64,000$ tokens. Note that, the batch sizes for training NAT is typically larger than the AT model, which improves final results. There are also specialized training settings for each models:

- **iNAT** (Lee et al., 2018): following the original paper, we train the iNAT model jointly with $4$ iterations of refinement during training. For each iteration, the model has the $50\%$ probability to learn as a denoising autoencoder, and the rest of the probability to learn from the model's own prediction.

- **InsT** (Stern et al., 2019): in this work, we only consider training the Insertion Transformer (InsT) using the slot-loss based on the uniform loss function (Stern et al., 2019). That is, we assign equal probabilities to all the insertable tokens inside each slot.

- **MaskT** (Ghazvininejad et al., 2019): following the original paper, we train the model as a typical masked language model where the ratio of masked tokens is sampled from $0 \sim 100\%$.

---

[6] `https://github.com/pytorch/fairseq/blob/master/examples/translation`.

- **LevT** (Gu et al., 2019): in this work, we only consider sequence generation tasks, which means the training of LevT is very similar to InsT. We use sentences with randomly deleted tokens to learn insertion, and learn deletion based on the model's own prediction.

**Decoding** For a fair comparison over all the NAT models, we use greedy decoding for all the models without considering any advanced decoding methods such as searching or re-ranking from a teacher model. For the vanilla NAT and FlowSeq, decoding is quite straight-forward and simply picks the $\arg\max$ at every position. For iNAT and MaskT, we fix the decoding steps to $10$. Both InsT and LevT decode in an adaptive number of iterations, and we set the maximum iterations for both models to be $10$. A special EOS penalty that penalizes generating too short sequences is tuned based on the validation set for both InsT and LevT.

For all models, final results are calculated using tokenized BLEU score.

## B   REAL DATA STATISTICS

The detailed dataset split for WMT14 En-De is shown in Tab. 6. In Fig. 7, we also plot the histogram of the conditional entropy of each pair of sentences $\mathcal{H}(\boldsymbol{y}|\boldsymbol{x})$ in the real parallel data and different distilled data sets from the big-AT, base-AT, small-AT and tiny-AT respectively. It shows that the distribution of the sentence-level conditional entropy differs widely. The mode of $\mathcal{H}(\boldsymbol{y}|\boldsymbol{x})$ in the real data is the highest and follows by distilled data from the big-AT, base-AT, small-AT and tiny-AT. This observation aligns with the complexity value $C(d)$ proposed in §3.2.

| Dataset | Train | Valid | Test | Vocabulary |
|---|---|---|---|---|
| WMT'14 En-De | 4,500,966 | 3000 | 3003 | 37,009 |

Table 6: Dataset statistics for WMT14 En-De.

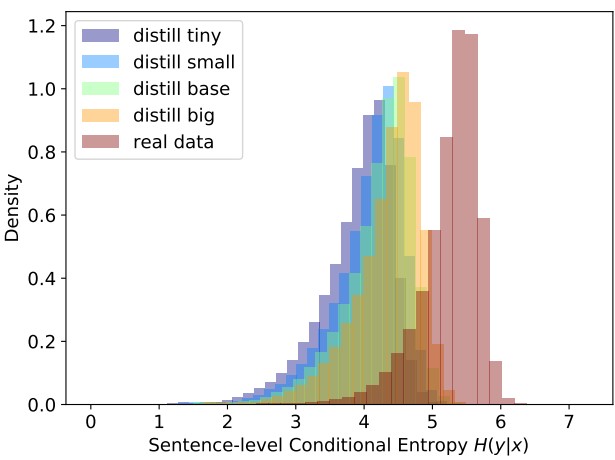

Figure 7: Density of conditional entropy $C(d)$ of each sentence pairs in different distilled data sets and the real data.

## C   ADDITIONAL METRICS

In Figure 8, we also showed results with different metrics together with BLEU scores considering that BLEU scores sometimes cannot fully capture the changes in the system. We considered 5 additional metrics in our experiments: METEOR (Banerjee & Lavie, 2005), RIBES (Isozaki et al., 2010), ChrF (Popović, 2015) TER (Snover et al., 2006), and BEER (Stanojevic & Simaan, 2014). Not surprisingly, we find that all the metrics are correlated with the original BLEU scores quite well showing a similar trend as discussed earlier.

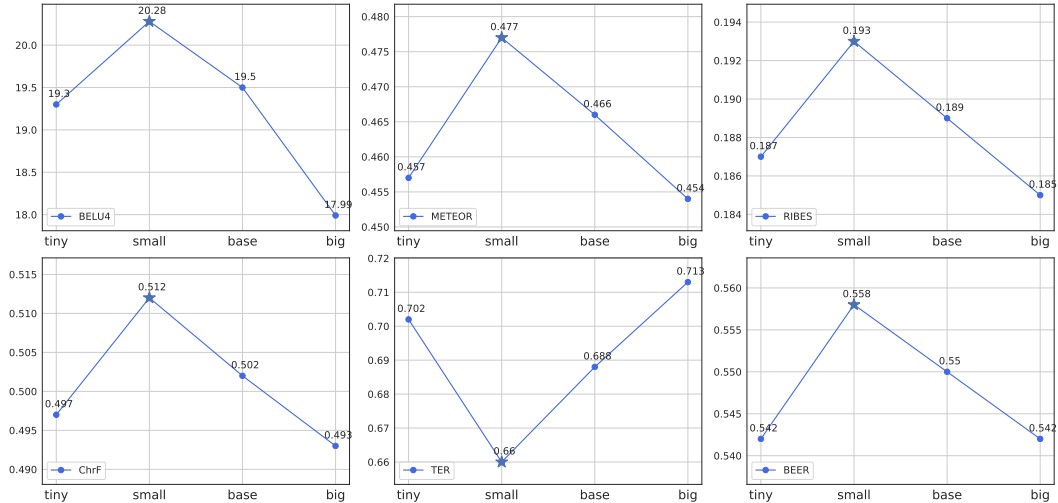

Figure 8: The performance of variant measure (BLEU ↑, METEOR ↑, RIBES ↑, ChrF ↑, TER ↓, BEER ↑) for the vanilla NAT model trained on the distilled data from tiny, small, base and big AT models on WMT14-ENDE newstest 2014 test sets.

## D  SYNTHETIC DATA WITH ACCESS TO THE TRUE DISTRIBUTION

### D.1  BACKGROUND: BAYESIAN DECISION THEORY

Bayesian decision theory is a fundamental statistical approach to the problem of pattern classification, which provides a principled rule of finding the optimal classification decision using probability and losses that accompany such decisions.

In the problem of structured prediction (Ma et al., 2017), let $x$ denote the input sequence and $y$ denote the output label sequence. Let $\mathcal{H}$ denote all the possible hypothesis functions from the input to the output space: $\mathcal{H} = \{h : \mathcal{X} \rightarrow \mathcal{Y}\}$. Let $r(y|x)$ denote the conditional risk on the input $x$, which is the expected loss of predicting $y$ based on the posterior probabilities:

$$r(y|x) = \mathbb{E}_{P(y'|x)}[L(y, y')], \tag{5}$$

, where $L(y, y')$ is the loss function that penalizes predicting the true target $y'$ as $y$. The classification task aims to find a hypothesis function $h$ that minimizes the overall risk $R$ given by

$$R(h) = \mathbb{E}_{P(x)}[r(h(x)|x)] \tag{6}$$

This is known as the Bayes risk. To minimize the overall risk, obviously we need to minimize the conditional risk for each input $x$. The Bayesian decision rule states that the global minimum of $R(h)$ is achieved when the classifier make predictions that minimize each conditional risk given $x$ and this gives the Bayes optimal classifier:

$$h^*(x) = \arg\min_{y \in \mathcal{Y}} r(y|x) \tag{7}$$

Let us consider two loss functions defined in Eq. 5. First is the sequence-level loss $L_{seq}(y, y') = 1 - \mathbb{I}(y = y')$, then in this case the Bayes classifier is:

$$h^*_{seq}(x) = \arg\max_{y \in \mathcal{Y}} P(y|x) \tag{8}$$

, which is the most probable output label sequence given the input sequence $x$.

Second let us consider the token-level loss $L_{tok}(\boldsymbol{y}, \boldsymbol{y}') = \sum_{t=1}^{T} 1 - \mathbb{I}(y_t = y_t')$, i.e the sum of zero-one loss at each time step. We have:

$$
\begin{aligned}
h_{tok}^*(\boldsymbol{x}) &= \underset{\boldsymbol{y} \in \mathcal{Y}}{\arg\min} \, \mathbb{E}_{P(\boldsymbol{y}'|\boldsymbol{x})}[L_2(\boldsymbol{y}, \boldsymbol{y}')] \\
&= \underset{\boldsymbol{y} \in \mathcal{Y}}{\arg\max} \, \mathbb{E}_{P(\boldsymbol{y}'|\boldsymbol{x})}[\textstyle\sum_{t=1}^{T} \mathbb{I}(y_t = y_t')] \\
&= \underset{\boldsymbol{y} \in \mathcal{Y}}{\arg\max} \, \textstyle\sum_{t=1}^{T} \mathbb{E}_{P(\boldsymbol{y}'|\boldsymbol{x})}[\mathbb{I}(y_t = y_t')] \\
&= \underset{\boldsymbol{y} \in \mathcal{Y}}{\arg\max} \, \textstyle\sum_{t=1}^{T} \mathbb{E}_{P(y_t'|\boldsymbol{x})}[\mathbb{I}(y_t = y_t')] \\
&= \underset{\boldsymbol{y} \in \mathcal{Y}}{\arg\max} \, \prod_{t=1}^{T} P(y_t|\boldsymbol{x})
\end{aligned}
\tag{9}
$$

This suggests that the Bayes classifier finds the most probable label at each time step given the input sequence.

## D.2 Experimental Setups and Analysis

To study how training data affects the performance of a weaker classifier, we construct a Hidden Markov Model (HMM) by sampling the parameters of the transition and emission probabilities uniformly within $(0, a]$ and $(0, b]$ respectively. A higher value of $a$ and $b$ indicates an HMM model with higher uncertainty. We refer this HMM as the "true HMM" as our real data generator. Next we consider a weaker classifier that uses a low-dimension bidirectional-LSTM (Bi-LSTM) to encode the input sequence and individual softmax functions at each time step to predict labels independently, which is referred as the "Bi-LSTM" classifier. Obviously, the Bi-LSTM classifier is not able to model the dependencies between output labels embedded in the HMM, and it is equivalent to a simplified non-autoregressive generation model.

We generate the real training data $D_{real} = \{(\boldsymbol{x}_1, \boldsymbol{y}_1), \cdots, (\boldsymbol{x}_N, \boldsymbol{y}_N)\}$ of size $N$ by sampling from the joint probability of the true HMM. Similarly we sample $N_{test}$ data points as the test data and $N_{valid}$ data points as the validation data. We evaluate the classifier's token-level accuracy $tacc$ and sequence-level accuracy $sacc$ on the test data respectively, where $tacc = \frac{\sum_{i=1}^{N_{test}} \sum_{t=1}^{T} \mathbb{I}(h(\boldsymbol{x}_i)^t = \boldsymbol{y}_i^t)}{T \times N_{test}}$ and $sacc = \frac{\sum_{i=1}^{N_{test}} \mathbb{I}(h(\boldsymbol{x}_i) = \boldsymbol{y}_i)}{N_{test}}$. These two metrics correspond to the token-level loss $L_{tok}$ and sequence-level loss $L_{seq}$ on each data point of the test data.

First, we use $h_{seq}^*(\boldsymbol{x})$ to generate the distillation labels $\boldsymbol{y}'$ from the true HMM, which corresponds to applying the Viterbi decoding to each $\boldsymbol{x}_i$ in $D_{real}$. The training data set $D_{seq}$ is created with $(\boldsymbol{x}_i, \boldsymbol{y}_i')$. Next, we use $h_{tok}^*(\boldsymbol{x})$ to generate the distillation labels $\hat{\boldsymbol{y}}$ and create the training data $D_{tok}$ of $(\boldsymbol{x}_i, \hat{\boldsymbol{y}}_i)$. To generate $\hat{\boldsymbol{y}}$, we apply the forward-backward algorithm to each $\boldsymbol{x}_i$ in $D_{real}$ and obtain $P(y_i^t|\boldsymbol{x}_i)$. We take $\arg\max$ over the label space $\mathcal{L}$: $\hat{y}_i^t = \underset{y_i^t \in \mathcal{L}}{\arg\max} \, P(y_i^t|\boldsymbol{x}_i)$.

We use these three training data ($D_{real}, D_{tok}, D_{seq}$) to train the Bi-LSTM classifier respectively. We repeat the experiment for 50 times by constructing 50 HMM models with different random seeds as the data generator. We find that when evaluating with the token-level accuracy $tacc$, models trained with $D_{tok}$ yields the best performance (Bi-LSTM trained with $D_{tok}$ win 97.6% runs); when evaluating with the sequence-level accuracy $sacc$, models trained with $D_{seq}$ yields the best performance (Bi-LSTM trained with $D_{seq}$ win 98.5% runs). This is because the Bi-LSTM classifier has difficulty modeling the true data distribution defined by an HMM. On the other hand, it is easier for the Bi-LSTM classifier to model the distributions of $D_{seq}$ and $D_{tok}$. Data sets $D_{seq}$ and $D_{tok}$ define deterministic conditional distributions over the input data, which are much simpler than the real data distribution. By definition, $D_{tok}$ is created by the optimal Bayes classifier $h_{tok}^*(\boldsymbol{x})$, this means that the Bi-LSTM classifier trained with $D_{tok}$ can better capture the distribution of $P(y_t|\boldsymbol{x}) = \underset{u_t}{\max} \, P(u_t|\boldsymbol{x})$, which can generalize better to the test data when evaluated with the token-level accuracy. Similarly, Bi-LSTM trained with $D_{seq}$ performs better on the test data with the sequence-level metric.

This corroborates our observation in machine translation task that NAT has difficulty in modeling the real conditional distribution of true sentence pairs. However, when using the distilled data translated

from a pretrained autoregressive model with beam-search decoding, it performs better on the test set when evaluated with the BLEU score metric.

