# OpenReview forum: "Understanding Knowledge Distillation in Non-autoregressive Machine Translation"
_ICLR.cc/2020/Conference — Accept (Poster)_

### Official Review · AnonReviewer3 · 2019-10-05
**Official Blind Review #3**

**Rating:** 8

**Review:**

[EDIT: Thanks for the response! I am updating my score to 8 given the rebuttal to my and other reviewers' questions]

--------------------------------------------
Summary:
--------------------------------------------
This paper studies knowledge distillation in the context of non-autoregressive translation. In particular, it is well known that in order to make NAT competitive with AT, one needs to train the NAT system on a distilled dataset from the teacher model. Using initial experiments on EN=>ES/FR/DE, the authors argue that this necessity arises from the overly-multimodal nature of the output distribution, and that the AT teacher model produces a less multimodal distribution that is easier to model with NAT.

Based on this, the authors propose two quantities that estimate the complexity (conditional entropy) and faithfulness (cross entropy vs real data), and derive approximations to these based on independence assumptions and an alignment model.  The translations from the teacher output are indeed found to be less complex, thereby facilitating easier training for the NAT student model.

--------------------------------------------
Strengths:
--------------------------------------------
- Careful initial experiments to motivate the study.

- The metrics under consideration (entropy/faithfulness) are carefully derived, and while some of the assumptions may be overly simplifying (e.g. conditional independence assumptions), they are reasonable enough such that the directionality is likely to be correct.

- Thorough comparison of various NAT models. This will benefit the community greatly since to my knowledge, there has been no empirical head-to-head comparison of the different NAT methods that exist today.

- Extensive experiments across various settings, e.g. varying the teacher model size, varying the decoding strategies, investigating BAN/MoE/Sequence-level interpolation, etc.

--------------------------------------------
Weaknesses:
--------------------------------------------
- It would have been interesting to consider a synthetic data setting such that one has access to the true underlying data distribution, such that approximations are not necessary.

- While translation is an important application of non-autoregressive generation, it would have also been interesting to study this in other seq2seq regimes such as summarization where the conditional entropy would presumably be even higher. (However this would complicate things like calculation of alignment probabilities, etc.)

--------------------------------------------
Other Questions/Comments:
--------------------------------------------
- p(y_t | l_i) coming from a token frequency distribution seems a bit too simple. Do the plots change if you model p(y | l_i) with a full language model that conditions on l_i?

- It's interesting to note that in Figures 1b,1c,1d, there is much more overlap between the romance languages es/fr, which are more closely related to each other.

- I am not sure I agree with "C(d) reflects the level of multi-modality of a parallel corpus". One can certainly imagine a distribution which is unimodal but has high entropy...

- It seems like that we want a teacher model with low complexity and high faithfulness. Have the authors tried training a teacher model to directly target this? The usual MLE objective obviously targets faithfulness, but perhaps one could use RL techniques to optimize for low complexity as well.



**Experience Assessment:**

I have published one or two papers in this area.

**Review Assessment: Checking Correctness Of Derivations And Theory:**

I assessed the sensibility of the derivations and theory.

**Review Assessment: Checking Correctness Of Experiments:**

I carefully checked the experiments.

**Review Assessment: Thoroughness In Paper Reading:**

I read the paper thoroughly.

---

> ### Author Response · Authors · 2019-11-13
> **Response to Reviewer #3**
>
> Thanks for your insightful comments! We address your concerns and questions as follows:
>
> - Weakness 1 [Synthetic data with known data distribution]:
> We have conducted experiments with synthetic data generated from a hidden markov model (HMM) and due to space limitations, we didn’t include it in the main paper.
>  - In our revised version, we have added this case study in the Appendix C.
>  - Tl; dr: we use constructed HMM as our data generator and train a bi-LSTM sequence labeling model (independent predictions). We find that the best distilled data from the HMM model correlates with the evaluation metric of sequence labeling (derived from the Bayesian decision theory), i.e. the best distilled data for sequence-level accuracy is generated by finding the most probable sequence out of the HMM model: $argmax_{y \in \mathcal{Y}} P(y|x)$ and the best distilled data for token-level accuracy is generated by finding the most probable label at each time step: $argmax_{y \in \mathcal{Y}} \prod_{t=1}^{T} P(y_t|x)$.
>
> - Weakness 2 [Other Seq2Seq tasks]:
> To make the experiments throughout the paper more consistent, we focus on the machine translation task. However, we have actually conducted experiments on other sequence-to-sequence tasks as well including summarization and automatic post-editing and find that knowledge distillation is consistently a key ingredient to improve the performance of non-autoregressive generation. But we ran these as preliminary experiments, we don’t have numbers now and will consider adding them to our final version.
>
> - Question [Do the plots change if you model p(y | l_i) with a full language model that conditions on l_i?]:
> In our initial experiments, we also tried to model the sentence probability instead of the average of tokens in the figure. However, since the sentence probability is basically a chain product of token level probabilities, it is very sensitive to small values for the less frequent words resulting in close to 0 values.
>
> - Question [Have the authors tried training a teacher model to directly target at the complexity and faithfulness?]:
> We haven’t tried this yet, but will try to resolve it in our future work. One of our major concerns is that the complexity of a data set is defined on all data samples, how to directly optimize the complexity over a data set for the teacher model is not easy at first glance. Batch-level learning and RL might be a reasonable solution.

---

### Official Review · AnonReviewer2 · 2019-10-22
**Official Blind Review #2**

**Rating:** 3

**Review:**

The paper analyses recent distillation techniques for non-autoregressive machine translation models (NAT). These models use a autoregressive teacher (AT), which typically perform better. However, AT models can not be parallelized that easily such as the NAT models. The distillation has the effect of removing modes from the dataset which helps the NAT models as they suffer from the averaging effect of maximum likelihood solutions. The authors analyze why such distillation is needed and what the effect of the complexity of the training set is and further propose 3 methods to adjust the complexity of the teacher to the complexity of the NAT model.

The paper is well written and easy to understand and the experiments are exhaustive and well done. I think the analysis using the synthetic dataset is nice, but I am not sure how surprising the results are. I think most of it has to be expected given the properties of maximum likelihood estimation. Hence, I think the contribution of the paper is a bit limited. I am, however, not an expert in the field to fully judge the contribution and will therefore put low confidence on my review.

**Experience Assessment:**

I do not know much about this area.

**Review Assessment: Checking Correctness Of Derivations And Theory:**

I did not assess the derivations or theory.

**Review Assessment: Checking Correctness Of Experiments:**

I assessed the sensibility of the experiments.

**Review Assessment: Thoroughness In Paper Reading:**

I read the paper at least twice and used my best judgement in assessing the paper.

---

> ### Author Response · Authors · 2019-11-13
> **Understanding Knowledge Distillation is Important to Non-autoregressive Generation**
>
> Thanks for your reviews!
> Though the results of the synthetic experiment can be expected given the properties of MLE, this experiment is only used as an introductory and explicit example to motivate our study of (1) how it is hard for a non-autoregressive student model (NAT) to capture modes by training with MLE; (2) how the distillation data set from an autoregressive teacher model (AT) removes the modes and can affect the performance of NAT. We are interested in understanding to what extent, and why, distillation helps in the learning of NAT models which has not been explained in prior works.
>
> As the reviewer has indicated that they are not very familiar with the area, we would just like to re-iterate that knowledge distillation is both (1) very important to building better NAT models (used in all state-of-the-art models), but (2) at the same time almost entirely glazed over in previous work both theoretically and empirically.
>
> Our work contributes by both improving understanding, and using this understanding to improve final results, sometimes by a great margin:
> - We first systematically studied why and how (mode reduction) non-autoregressive generation models benefit from the knowledge distillation technique. We then propose two metrics to measure the complexity and faithfulness of a given training data set and show how they correlate with the performance of an NAT model.
> - We conduct extensive experiments over different AT teacher models and NAT student models to reveal the correlation between the capacity of an NAT model and the optimal (in terms of the complexity) distillation data from the AT model.
> - We further propose several techniques that can be used to adjust the complexity of the distilled data to match the student model’s capacity. We have achieved state-of-the-art performance for NAT and almost closes the gap between NAT and AT models.

---

### Official Review · AnonReviewer5 · 2019-11-13
**Official Blind Review #5**

**Rating:** 8

**Review:**

In this paper, the authors investigate non-autoregressive translation (NAT). They specifically look into how using different auto-regressive translation (AT) models for knowledge distillation impacts the quality of NAT models. The paper is well organised and the experiments are sound and interesting, shedding light on an aspect of NAT models that's been rather dismissed as secondary until now: the impact of varying AT knowledge distillation.

First, so as to better please those out there who are more "state-of-the-art" inclined, I suggest the authors to better emphasise their improvements. Results obtained by their analysis can lead to improvements as stated in the last paragraph of Section 4. This could be better stressed in the introduction and it would make the main take-away messages from the paper stronger.

On a more general note, I would like to know how robust these models are. I understand this is a problem NAT and machine translation papers have in general, but I would still like to suggest the authors train each model from scratch multiple times using different random seeds, and report mean and variance of their results (i.e. BLEU). Although I understand that training each AT model 4 times and each NAT model 4x4 times (multiple times for each AT model trained) is unfeasible, you could still report mean and variance separately for AT and NAT, and simply choose one out of the 4 trained AT models per architecture to perform distillation. I recommend training each model at least 3 times and reporting mean BLEU and variance.

I would also recommend using other MT metrics (e.g. chrF, METEOR, BEER, etc.), since BLEU is a comparatively weak metric in terms of correlations with human judgements. For translations into German, look into characTER, chrF and BEER. For more information on the different metrics available, how they correlate with human judgements, and which better apply to different target languages / language pairs, please refer to the WMT evaluation/metrics shared tasks. [1]

I have a few general comments and suggestions:
- In Section 3.1, when introducing the NAT model (simplified version of Gu et al., (2019)), be more specific. I would like to see an actual description of what the modifications consist of in more detail: is there a sentence length prediction step? what happens when source length is different from source length? etc.
- In Section 3.2, right after Equation (3), the authors refer to "x and y" in an inverted manner, please fix it.
- In Section 4.3, you mention that two of the decoding methods used involve "random sampling", which I find misleading. You probably mean sampling according to the model distribution p(y_t | y_{<t}) for all t, which is not random. I suggest you simply remove the word "random" and mention that you use "sampling" and "sampling within the top-10 candidates". Also, when you sampling using the top-10 candidates, do you simply re-normalise the probability mass to include only the top-10 candidate tokens?
- As a suggestion, Tables and Figures could be more self-contained. There is always a compromise between conciseness and clarity, added by the fact that the page limit makes things even harder. However, I would recommend including more information in Table/Figure captions, especially in Figure 3 and Tables 3 and 4. Try to at least mention the training/evaluation data (dev or test?). Ideally one should understand it from reading the abstract and carefully reading the caption.
- In Appendix A, you mention that you select models according to validation loss. Is that really the case? If so, why? I am not sure whether validation loss (i.e. word cross-entropy on the validation set) should correlate so well with translation quality (or sentence-level metrics such as BLEU).

[1] http://www.statmt.org/wmt19/metrics-task.html

**Experience Assessment:**

I have published in this field for several years.

**Review Assessment: Checking Correctness Of Derivations And Theory:**

I assessed the sensibility of the derivations and theory.

**Review Assessment: Checking Correctness Of Experiments:**

I carefully checked the experiments.

**Review Assessment: Thoroughness In Paper Reading:**

I read the paper thoroughly.

---

> ### Author Response · Authors · 2019-11-15
> **Thank you for your comments! We update the paper accordingly.**
>
> Thanks for your insightful reviews and we appreciate your valuable suggestions!
> We have revised the paper and reflected most of your suggestions.
>
> We address your concerns and questions as follows:
> Question [sampling top-k decoding]: Yes, we renormalize the top-k decoding probabilities and do sampling.
> Question [model selection according to validation loss]: We used the default setting of fairseq (the framework we used for implementing the model) - valid loss for validation. For FlowSeq, we used the open source implementation which used valid BLEU to select model. We have revised the paper to make it more clear.
>
> In the latest revision, we also included figures of NAT model performance measured by other metrics (e.g. METEOR, TER, RIBES, ChrF, BEER), and we found that all the metrics are correlated with the original BLEU scores quite well showing the similar trend.

---

### Public Comment · ~Kaaliya_Budhil2 · 2019-10-11
**other language pairs**

Very complete and impressive work!

I have a question about language pairs. As shown by previous works such as the original NAT and mask-predict, knowledge distillation has more impact on WMT14 En-De/De-En. However, it conducts minor effects on WMT16 En-Ro/Ro-En. So I think the performance of knowledge distillation is related to specific language pairs. While you only analyze the translation from English to German, have you conducted experiments on other language pairs or directions (de-en)? Are the conclusions the same as that on en-de?

---

> ### Author Response · Authors · 2019-10-11
> **We have done experiments for other language pairs and the conclusion is the same.**
>
> Thanks for your comments!
>
> Actually, we have done experiments on other language pairs and directions, e.g. de-en, ro-en. We had the same conclusions for these different language pairs. Due to space limitation, we didn't include them in our paper.
>
> For example, for ro-en, we find that the conditional entropy C(d) of different distillation data from AT models of different capacity is very close to each other in contrast with the en-de case (shown in Figure 7 Appendix B.), and the gap between the real data and the distillation data is not as large as that of en-de as well. Accordingly, we observe that distilling from different AT models of different capacity doesn't show improvements especially for bigger-capacity NAT models, e.g. with LevT we get 32.22 vs 33.1 vs 33.26 with real data, small AT distillation and base AT distillation data respectively.
>
> We will definitely add experiments on more language pairs in the final version for completeness.

---

> > ### Public Comment · ~Kaaliya_Budhil2 · 2019-10-21
> > **fuzzy reordering score**
> >
> > Thanks a lot for your reply!
> >
> > A small question. I think the analysis in Section 4.2 is very interesting and instructive, could you provide more details about computing the fuzzy reordering score? I find that the cited paper (Talbot et.al., 2011) did not provide the source code.

---

> > > ### Author Response · Authors · 2019-10-21
> > > **Equation of fuzzy reordering score**
> > >
> > > Thanks for your interest!
> > > The fuzzy reordering score is defined as $FRD = 1 - \frac{C-1}{M-1}$, where $C$ is the number of chunks of contiguously aligned words and $M$ is the number of words in the source sentence, which is easy to compute with several lines of code. We will release code afterwards.

---

### Public Comment · ~Xuanli_He2 · 2020-01-30
**How to measure the faithfulness**

It's a very insightful study. I have one question related to the measure of faithfulness. From the paper, I couldn't figure out how you align x and y in your $P_r(y|x)$ and $P_d(y|x)$ in the equ (4).

And you might miss a related work He et al. 2018 [1] which suggested the diverse translation with MOE earlier than Shen et al. 2019.

[1] Sequence to Sequence Mixture Model for Diverse Machine Translation, He et al. 2018

---

> ### Author Response · Authors · 2020-01-30
> **Thanks for the comment!**
>
> Thanks for pointing out the reference!
>
> When we compute the alignment distribution for $p(y|x)$, we set the probabilities of target words y to be a very small value if it doesn't exist in the alignment distribution given a x.

---

> > ### Public Comment · ~Xuanli_He2 · 2020-01-31
> > **Still unclear for the alignment distribution**
> >
> > Sorry, let me elaborate on my words. In equation (4), $y$s and $x$s are target words and source words respectively. My question here is how you align the source word $x$ to the target word $y$, did you also use fast_align or any other toolkit? or according to the attention scores? After you finished the alignments, how could you compute the alignment distribution $p(y|x)$

---

> > > ### Author Response · Authors · 2020-02-01
> > > **We use fast_align**
> > >
> > > Yes, you are right, we use fast align to compute the alignments, it basically outputs the target tokens that each source token is aligned, and then we can just count and normalize to compute the alignment distribution.

---

> > > > ### Public Comment · ~Xuanli_He2 · 2020-02-02
> > > > **Thanks for your patience and answers**
> > > >
> > > > My problem is addressed. Thanks for your time.

---

### Decision · Program_Chairs · 2019-12-19

**Decision:**

Accept (Poster)

**Comment:**

Main content:

Blind review #3 summarized it well, as follows:

This paper studies knowledge distillation in the context of non-autoregressive translation. In particular, it is well known that in order to make NAT competitive with AT, one needs to train the NAT system on a distilled dataset from the teacher model. Using initial experiments on EN=>ES/FR/DE, the authors argue that this necessity arises from the overly-multimodal nature of the output distribution, and that the AT teacher model produces a less multimodal distribution that is easier to model with NAT.

Based on this, the authors propose two quantities that estimate the complexity (conditional entropy) and faithfulness (cross entropy vs real data), and derive approximations to these based on independence assumptions and an alignment model.  The translations from the teacher output are indeed found to be less complex, thereby facilitating easier training for the NAT student model.

--

Discussion:

Questions were mostly about how robust the results were on other language pairs and random starting points. Authors addressed questions reasonably.

One low review came from a reviewer who admitted not knowing the field, and I agree with the other two reviewers.

--

Recommendation and justification:

I think papers that offer empirically support for scientific insight (giving an "a-ha!" reaction), rather than massive engineering efforts to beat the state of the art, are very worthwhile in scientific conferences. This paper meets that criteria for acceptance.